# Secondary Attack Rate among Non-Spousal Household Contacts of Coronavirus Disease 2019 in Tsuchiura, Japan, August 2020–February 2021

**DOI:** 10.3390/ijerph18178921

**Published:** 2021-08-25

**Authors:** Tsuyoshi Ogata, Fujiko Irie, Eiko Ogawa, Shifuko Ujiie, Aina Seki, Koji Wada, Hideo Tanaka

**Affiliations:** 1Itako Public Health Center of Ibaraki Prefectural Government, Itako 311-2422, Japan; 2Tsuchiura Public Health Center of Ibaraki Prefectural Government, Tsuchiura 300-0812, Japan; f.irie@pref.ibaraki.lg.jp (F.I.); hi.ogawa@pref.ibaraki.lg.jp (E.O.); ai.seki@pref.ibaraki.lg.jp (A.S.); 3Tsukuba Public Health Center of Ibaraki Prefectural Government, Tsukuba 305-0035, Japan; fuji1981425@hotmail.com; 4Department of Public Health, Faculty of Medicine and Graduate School of Public Health, International University of Health and Welfare, Tokyo 107-8402, Japan; kwada@iuhw.ac.jp; 5Fujiidera Public Health Center of Osaka Prefectural Government, Fujiidera 583-0024, Japan; TanakaH61@mbox.pref.osaka.lg.jp

**Keywords:** COVID-19, household transmission, secondary attack rate, non-spouse, household size, diagnostic delay, Japan

## Abstract

Household secondary attack rate (HSAR) by risk factor might have a higher transmission rate between spouses. We investigated risk factors for the HSAR among non-spousal household contacts of patients with coronavirus disease 2019 (COVID-19). We studied household contacts of index cases of COVID-19 in Tsuchiura, Japan, from August 2020 through February 2021. The HSARs of the whole household contacts and non-spousal household contacts were calculated and compared across risk factors. We used a generalized linear mixed regression model for multivariate analysis. We enrolled 496 household contacts of 236 index COVID-19 cases. The HSAR was higher for spousal household contacts (37.8%) than for other contacts (21.2%). The HSAR was lower for non-spousal household contacts with a household size (number of household members) of two (18.2%), compared to the HSAR for contacts with a household size ≥4. The HSAR was higher for non-spousal household contacts of index patients with ≥3 days of diagnostic delay (period between onset and diagnosis) (26.0%) compared to those with ≤2 days’ delay (12.5%) (*p* = 0.033). Among non-spousal household contacts, the HSAR was low for those with a household size of two and was high for contacts of index patients with a long diagnostic delay.

## 1. Introduction

An outbreak of the novel coronavirus disease 2019 (COVID-19) began in December 2019 in China. It spread to other countries, including Japan, and the World Health Organization declared it a public health emergency of international concern [1].

The virus is transmitted through droplets, airborne, or contact with secretion from an infected individual [2,3]. In Wuhan, 2020, the mean incubation period of COVID-19 was 5.2 days, and the basic reproductive number was 2.2 [4]. COVID-19 patients transmit the virus to contacts from 2 days prior to the onset of symptoms and the median serial interval of COVID-19 was reported as 4–5 days [5,6]. Wearing a mask and keeping physical distance can decrease transmission [7,8,9]. Closed space with poor ventilation and eating at restaurants may increase transmission [7,10].

In Japan, COVID-19 patients at the earliest stage had been exposed to the virus in China, or on a cruise ship (Diamond Princess) quarantined off the Yokohama port in Japan. However, an increase in the number of COVID-19 patients who had not visited China was reported in February 2020, and Japan experienced the first wave of COVID-19 with peak in April 2020 [11,12,13]. As of August 2021, Japan has had a surge of COVID-19 five times: the second wave with peak in August 2020, the third wave with peak in January 2021, the fourth wave with alpha variant and peak in May 2021, and the fifth wave with delta variant ongoing. A total of 1.2 million COVID-19 patients were reported in Japan [14]. Government of Japan has requested people to keep physical distance and wear mask outside the household, but not necessarily in the household [15].

The secondary attack rates of household contacts for COVID-19 are significant for assessing the transmissibility of severe acute respiratory syndrome coronavirus 2 (SARS-CoV-2), the virus causing COVID-19, and risk factors for infectivity of index patients and susceptibility of contacts [16]. Studies regarding the household secondary attack rate (HSAR) of COVID-19 have been published in various countries.

A meta-analysis reported the estimated HSAR to be 16.6%, and it increased for spousal household contacts [16]. Several other previous studies also reported higher HSARs for spouses [17,18,19,20,21,22]. 

The meta-analysis also reported risk factors to be contacts who were older, contacts in households with a size of two (having only one household contact) [16]. However, HSAR by risk factors, such as size of household or age, might be influenced by spousal relationship, with a higher transmission rate between spouses. Therefore, analysis of the HSAR for household contacts other than spouses may be necessary [16]. To our knowledge, there are no studies analyzing HSAR strictly between non-spousal household contacts. 

This study specifically addressed several questions: firstly, several previous studies reported higher HSARs for contacts with a household size of two compared to those of contacts with household size ≥4, and contacts with older age [23,24,25]. The present study tried to determine whether the HSAR for contacts with a small household size or older age was high only because it was confounded by a spousal relationship or whether it was also high even when contacts were limited to a non-spouse [16]. 

Secondly, a previous study reported a higher HSAR for contacts of index patients with a long diagnostic delay [26]. However, a spousal contact might spend a greater amount of time in the same room with the index case compared to a non-spousal contact, and the impact of diagnostic delay might differ between spousal and non-spousal contacts. The present study aimed to determine whether the HSAR for contacts of index patients with long diagnostic delay was also high even when contacts were limited to a non-spouse. 

In summary, this study aimed to elucidate HSARs by risk factors among non-spousal household contacts of patients with COVID-19.

## 2. Materials and Methods

### 2.1. Study Design

The study used a cross-sectional study design.

### 2.2. Setting

This study was carried out in the jurisdiction of the Tsuchiura Public Health Center of the Ibaraki Prefectural Government in Japan, which included three cities—Tsuchiura City, Kasumigaura City, and Ishioka City—of Ibaraki Prefecture. The public health center serves a population of approximately 250,000. The area is located about 80 km northeast of Tokyo and has easy access to Tokyo by railroads and highways.

### 2.3. Index COVID-19 Cases

The index COVID-19 cases eligible for this study involved individuals living in the jurisdiction with confirmed SARS-CoV-2 infection, as defined by the Tsuchiura Public Health Center, from August 2020 through February 2021. The number of confirmed patients with COVID-19 living in the jurisdiction of the Tsuchiura Public Health Center was 16 at the end of August 2020 and 638 at the end of February 2021. None were vaccinated against COVID-19, and no cases of a virus variant of concern had been detected in Ibaraki until February 2021. The first cases of virus variant were detected in the twelfth week (22–28 March 2012) in Ibaraki, and infection of these cases were confirmed in March [27].

In Japan, according to the Infectious Diseases Control Law (The Law), the public health center must be notified of all COVID-19 cases [3]. SARS-CoV-2 infections were confirmed using polymerase chain reaction (PCR) tests with a cycle threshold value of 40, loop-mediated isothermal amplification tests, antigen quantitative tests, or monoclonal antigen qualitative tests. The PCR test was implemented if the results of any of the other tests were not definite. 

The public health center implemented an epidemiological investigation of the patients based on the law. The public health center nurses interviewed the patients and collected data on demographics, symptoms, and history of a definite contact with a patient with COVID-19. 

We defined patients with COVID-19 with apparent exposure to SARS-CoV-2 as the index case in a household. If no patient with COVID-19 had exposure to SARS-CoV-2 and many patients with COVID-19 in a household had symptoms, the patient with COVID-19 who had the earliest onset date was defined as the index case in a household, and the other members in the household were included as participant contacts. If a household had two members with the same earliest onset date, it was excluded from the analysis.

### 2.4. Participant Contacts

The participants of this study were household contacts of index patients with COVID-19, living with the patient and usually sleeping in the same house. If an index case had no household contact, the household was excluded from analysis. As the number of confirmed COVID-19 cases per population in the jurisdiction of the Tsuchiura Public Health Center was 0.26% at the end of February 2021, we assumed that household contacts were susceptible to SARS-CoV-2 infection.

The public health center implements a law-based bidirectional contact tracing of the patients, whether symptomatic or not [28]. Based on the regulations on infectious diseases, Tsuchiura Public Health Center collected PCR test samples on all the household contacts of index cases. If a contact had a negative PCR test result but had new symptom onset, we implemented another PCR test. 

The PCR test was basically implemented at the Institute of Health of Ibaraki prefectural government. If household contacts had undergone a SARS-CoV-2 test at an institute other than the Institute of Health of Ibaraki prefectural government because of convenience, the household was excluded owing to difficulties with obtaining informed consent.

### 2.5. Outcome, Data Collection, and Variables

The outcome of interest in the study was SARS-CoV-2 transmission to household contacts of index COVID-19 cases. 

Household contacts were interviewed by public health nurses. Through bidirectional contact tracing after SARS-CoV-2 confirmation, physicians and public health nurses of the Tsuchiura Public Health Center collected data on the size of the household, i.e., number of household members, the participants’ demographic data, date of symptom onset, and behaviors prior to testing [28]. 

### 2.6. Statistical Analysis

We described the characteristics of index patients with COVID-19 and household contacts. The secondary attack rates in the entire household contacts, non-spousal household contacts, and spousal household contacts were independently calculated, and compared across risk factors in index cases or household contacts. Data were presented as proportions with percentages and with 95% confidence intervals (CI). For multivariate analysis of the entire household contacts and non-spousal household contacts, we used a generalized linear mixed regression model to adjust for confounding by household cluster and calculated the adjusted odds ratio (aOR) and 95% CI.

Statistical analyses were performed using R (version 3.6-2; The R Foundation for Statistical Computing, Vienna, Austria).

### 2.7. Ethical Concerns

Written informed consent was obtained from the patients or their parents for study. The study was conducted in accordance with the recommendations outlined in the Declaration of Helsinki. The study protocol was approved on 8 July 2021 by the Ibaraki Prefecture Epidemiological Research Joint Ethics Review Committee (protocol number: R3-1). 

## 3. Results

We enrolled 236 index patients with COVID-19 and 496 household contacts (Table 1). The mean (standard deviation) age was 41.3 (19.3) years for index COVID-19 cases and 40.6 (22.9) years for household contacts. The median (interquartile) size of the households was 4 (3–5). The median (interquartile) diagnostic delay from onset was 3 (2–6) days.

In total, 119 (24.0%) contacts had spousal relationships with the index patients, and 89 (17.9%) contacts had a household size of two (Table 1). The remaining 377 non-spouses included 161 parents, 128 children, and 88 other contacts of the index patients.

Table 2 shows the prevalence of SARS-CoV-2 infection for household contacts. In total, 125 household contacts were infected with SARS-CoV-2: the overall HSAR was 25.2%.

The HSAR was higher for household contacts with spousal relationships to index COVID-19 patients (37.8%, 95% CI 30%–47%) compared to contacts with other relationships (21.2%, 95% CI 17%–26%) (aOR 2.85, 95% CI 1.25–6.5, *p* = 0.013). 

The HSAR was higher for household contacts with a household size of two (38.2%, 95% CI 29%–49%) compared to contacts with a household size ≥4 (21.5%, 95% CI 17%–27%). 

The HSAR was higher for household contacts of index patients with ≥3 days of diagnostic delay (30.2%, 95% CI 25%–36%) than for contacts of index cases with ≤2 days of diagnostic delay (17.6%, 95% CI 13%–24%). The HSAR was higher for household contacts of asymptomatic index cases (25.3%, 95% CI 18%–35%) than for contacts of index cases with ≤2 days of diagnostic delay.

Table 3 shows the prevalence of SARS-CoV-2 infection in 377 non-spousal household contacts; 80 (21.2%) contacts were infected.

The HSAR was lower for non-spousal household contact with a household size of two (18.2%, 95% CI 8%–35%) compared to contacts with a household size ≥4 (20.4%, 95% CI 16%–26%) (aOR 0.73, 95% CI 0.13–4.0). 

The HSAR was higher for non-spousal household contacts of index patients with ≥3 days of diagnostic delay (26.0%, 95% CI 20–33%) than for contacts of index cases with ≤2 days of diagnostic delay (12.5%, 95% CI 8–14%) (aOR 4.34, 95% CI 1.13–16.7, *p* = 0.033). The HSAR was not lower for household contacts of asymptomatic index cases (25.0%, 95% CI 16%–37%) compared to contacts of index cases with ≤2 days of diagnostic delay (aOR 2.45, 95% CI 0.47–13.8).

The HSAR was relatively high for non-spousal older household contacts aged ≥60 (28.8%) but not significantly higher than that for contacts aged 0–59 years (19.6%) (aOR 1.42, 95% CI 0.83–2.4).

Table 4 shows SARS-CoV-2 infection for 119 spousal household contacts.

The HSAR was not significantly higher for spousal household contacts of index patients with ≥3 days of diagnostic delay (42.6%, 95% CI 31%–55%) than for contacts of index cases with ≤2 days of diagnostic delay (38.7%, 95% CI 29%–50%) (aOR 1.01, 95% CI 0.37–2.6).

The HSAR for spousal contacts shown in Table 4 was generally higher than the HSAR for non-spousal contacts shown in Table 3 across various variables. For example, the HSAR for spousal contacts aged ≤59 was 36.0% (95% CI 27%–47%), and higher than that for non-spousal contacts (19.6%, 95% CI 16%–24%).

## 4. Discussion

The HSAR of COVID-19 was 25% in Tsuchiura, Japan, from August 2020 through February 2021. It was higher than that estimated by a previous meta-analysis [16]. During the study period, the cumulative incidence rate of COVID-19 was low, and participants were not influenced by the presence of a virus variant or vaccination.

The HSAR for contacts with a spousal relationship was 38%, which was higher than that of other contacts. This result was consistent with the findings of several previous studies reporting a higher HSAR for spouses [17,18,19,20,21,22]. In the present study, the higher HSAR for spousal contacts was consistent across various variables. The spousal contact may spend longer periods of time and eat or sleep more often in the same room with the index case compared to other household members.

In the present study, the HSAR was higher for the whole contacts with a household size of two compared to contacts with a household size ≥4. However, it was not higher when household contacts were limited to those other than a spouse. Several previous studies reported a higher HSAR for households with two contacts [23,24,25]. However, we did not find studies analyzing HSAR by household size only for non-spousal household contacts. From results of the present study, we assume that the high HSAR of contacts with a household size of two was due to the influence of a spousal relationship.

Several studies reported a higher HSAR for older people [23,24,25]. Older household contacts might stay longer in the house compared to other household contacts. In the present study, although HSAR was relatively high for non-spousal household contacts aged ≥60 years, it was not significantly higher than that of contacts aged <60 years. 

A previous study in Japan reported a higher HSAR for contacts of index patients aged 60–69 years [26]. However, in the present study, the HSAR was not higher for non-spousal household contacts of index patients aged ≥60 years.

The HSAR was 13% for household contacts of index patients with ≤2 days of diagnostic delay, which was significantly lower than the HSAR of contacts of index patients with ≥3 days of diagnostic delay. However, the HSAR was not significantly higher for spousal household contacts of index patients with a long diagnostic delay. In previous literature in Japan, the long diagnostic delay of COVID-19 was associated with a high HSAR for all household contacts [26]; it was also reported to correlate with subsequent community transmissions [29]. A study using a mathematical model showed that the delay between symptom onset and isolation played a major role in controlling the COVID-19 outbreak [30]. Self-quarantine of the index patient at symptom onset [18] or short effective contact duration [31] is beneficial for preventing household transmission of SARS-CoV-2. However, to the best of our knowledge, we did not find studies analyzing HSAR by diagnostic delay only for non-spousal household contacts. From results of the present study, we assume that it may not be easy for a spouse to prevent transmission despite a short diagnostic delay and that rigorous isolation from non-spousal household members after symptom onset of the index patient, as well as early diagnosis of COVID-19, may be especially significant interventions for preventing household transmission. 

Although a previous study reported a lower HSAR for contacts of asymptomatic index cases [23], the present study did not show such results.

The present study was implemented by the governmental body in charge of all the COVID-19 cases in the jurisdiction. The higher HSAR for spousal contacts across various variables might reflect the soundness of the study. The results suggested the size of household might not be important for transmission and intervention. They also suggested the importance of rigorous isolation of the index patient from non-spousal household members after symptom onset. However, they also suggested difficulties in preventing transmission to the spouse. Universal mask use between wife and husband in a house is an intervention worth discussing though it may not be easy to implement such a custom. Sufficient air ventilation such as opening house windows may be another important intervention.

This study had several limitations. First, this study used a cross-sectional design; thus, the results did not prove any causal relationships. Second, we basically performed the PCR test once for asymptomatic contacts, and we might have missed some asymptomatic infection of contacts. Third, we did not evaluate associated household environmental factors, including the level of crowding, lifestyle, and precaution measures of each contact, and proximity of contacts to the index cases. Fourth, we defined the patient with the earliest onset date as the index case in a household without any COVID-19 case with apparent exposure to SARS-CoV-2. There might be a possibility that the index cases might have been misclassified as secondary cases. Fifth, the study did not include variants of the virus. The results of the study are not necessarily externally validated for COVID-19 cases infected with variants of the virus.

Further studies are necessary to analyze the association between HSAR and risk factors, such as household size, age, diagnostic delay of the index patient, vaccination, and variant of the virus, considering household crowding [2], lifestyle, precaution measures, and proximity to the index case.

## 5. Conclusions

In assessing the risk factors affecting household contacts, it was useful to analyze HSAR for non-spousal contacts. The HSAR for a household size of two was not higher among non-spousal household contacts. However, it was higher for non-spousal household contacts of index patients with ≥3 days of diagnostic delay.

## Figures and Tables

**Table 1 ijerph-18-08921-t001:** Characteristics of COVID-19 patients and household contacts.

Variables	Index COVID-19 Cases	Household Contacts
N	236	496
Relationship to index patient		
Spouse		119 (24.0%)
Other		377 (76.0%)
The size of household		
2		89 (17.9%)
3		142 (28.6%)
≥4		265 (53.4%)
Diagnostic delay from onset		
≤2 days	73 (30.9%)	
≥3 days+	118 (50.0%)	
Asymptomatic	45 (19.1%)	
Sex		
Male	133 (56.4%)	224 (45.2%)
Female	103 (43.6%)	272 (54.8%)
Age		
≤59	190 (80.5%)	397 (80.0%)
≥60	46(19.5%)	99 (20.0%)
Diagnostic delay from onset		
≤2 days	73 (30.9%)	
≥3 days+	118 (50.0%)	
Asymptomatic	45 (19.1%)	

**Table 2 ijerph-18-08921-t002:** Secondary attack rate among household contacts.

Variables	Household Contacts	Infected Contacts	Secondary Attack Rate	Multivariate Analysis
			% (95% CI)	aOR (95% CI)
*n*	496	125	25.2 (21.6–29.2)	
Risk factors in household contacts				
Relationship to index patient				
Spouse	119	45	37.8 (29.6–46.8)	2.85 (1.25–6.5)
Other	377	80	21.2 (17.4–25.6)	1
The size of household				
2	89	34	38.2 (28.8–48.6)	2.05 (0.64–6.6)
3	142	34	23.9 (17.7–31.6)	1.04 (0.36–3.0)
≥4	265	57	21.5 (17.0–26.9)	1
Sex				
Male	224	54	24.1 (19.0–30.1)	1
Female	272	71	26.1 (21.2–31.7)	1.09 (0.57–2.1)
Age				
≤59	397	92	23.2 (19.3–27.6)	1
≥60	99	33	33.3 (24.8–43.1)	1.24 (0.78–1.9)
Risk factors in index COVID-19 cases				
Diagnostic delay from onset				
≤2 days	159	28	17.6 (12.5–24.4)	1
≥3 days	242	73	30.2 (24.7–36.2)	2.66 (0.95–7.5)
Asymptomatic	95	24	25.3 (17.6–34.9)	1.33 (0.36–4.9)
Sex				
Male	285	73	25.6 (20.9–31.0)	1
Female	211	52	24.6 (19.3–30.9)	0.90 (0.36–2.2)
Age				
≤59	429	100	23.3 (19.6–27.6)	1
≥60	67	25	37.3 (26.7–49.3)	1.08 (0.57–2.0)

All variables were included in the analysis. CI = confidence interval.

**Table 3 ijerph-18-08921-t003:** Secondary attack rate among non-spousal household contacts.

Variables	Household Contacts	Infected Contacts	Secondary Attack Rate	Multivariate Analysis
			% (95% CI)	aOR (95% CI)
*N*	377	80	21.2 (17.4–25.6)	
Risk factors in household contacts				
The size of household				
2	33	6	18.2 (8.4–34.9)	0.73 (0.13–4.0)
3	109	26	23.9 (16.8–32.7)	1.19 (0.36–4.0)
≥4	235	48	20.4 (15.8–26.1)	1
Sex				
Male	182	38	20.1 (15.6–27.4)	1
Female	195	42	21.5 (16.3–27.9)	1.23 (0.57–2.6)
Age				
≤59	311	61	19.6 (15.6–24.4)	1
≥60	66	19	28.8 (19.3–40.7)	1.42 (0.83–2.4)
Risk factors in index COVID-19 cases				
Diagnostic delay from onset				
≤2 days	128	16	12.5 (7.8–19.5)	1
≥3 days	181	47	26.0 (20.1–32.8)	4.34 (1.13–16.7)
Asymptomatic	68	17	25.0 (16.2–36.6)	2.45 (0.47–13.8)
Sex				
Male	210	44	21.0 (16.0–27.0)	1
Female	167	36	21.6 (16.0–28.5)	0.77 (0.25–2.3)
Age				
≤59	349	73	20.9 (17.0–25.5)	1
≥60	28	7	25.0 (12.5–43.7)	1.05 (0.42–2.6)

All variables were included in the analysis. CI = confidence interval.

**Table 4 ijerph-18-08921-t004:** Secondary attack rate among spouse household contacts.

Variables	Household Contacts	Infected Contacts	Secondary Attack Rate	Multivariate Analysis
			% (95% CI)	aOR (95% CI)
*N*	119	45	37.8 (29.6–46.8)	
Risk factors in household contacts				
The size of household				
2	56	28	50.0 (37.3–62.7)	2.01 (0.73–5.5)
3	33	8	24.2 (12.7–41.3)	0.66 (0.21–2.1)
≥4	30	9	30.0 (16.6–48.1)	1
Sex				
Male	42	16	38.1 (25.0–53.2)	1
Female	77	29	37.7 (27.7–48.9)	0.56 (0.08–3.9)
Age				
≤59	86	31	36.0 (26.7–46.6)	1
≥60	33	14	42.4 (27.3–59.2)	0.75 (0.37–1.5)
Risk factors in index COVID-19 cases				
Diagnostic delay from onset				
≤2 days	31	12	38.7 (23.8–56.2)	1
≥3 days	61	26	42.6 (31.0–55.1)	1.01 (0.37–2.6)
Asymptomatic	27	7	25.9 (13.1–45.0)	0.57 (0.18–1.9)
Sex				
Male	75	29	38.7 (28.5–50.0)	1
Female	44	16	36.4 (23.8–51.2)	0.53 (0.08–3.6)
Age				
≤59	80	27	33.8 (24.4–44.7)	1
≥60	39	18	46.2 (31.6–61.4)	1.49 (0.75–3.0)

All variables were included in the analysis. CI = confidence interval.

## Data Availability

The data presented in this study are available on reasonable request from the corresponding author. The data are not publicly available due to protection of personal information.

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
