# Peer review of "Secondary Attack Rate among Non-Spousal Household Contacts of Coronavirus Disease 2019 in Tsuchiura, Japan, August 2020–February 2021"

_ijerph, 2021, doi:10.3390/ijerph18178921_

Round 1

Reviewer 1 Report

As shown in the limitation, it was better to consider other factors such as preventive behaviors at home etc. in this study.

Overall, there is no problem with the research method.

About inequality signs in the table.
I think the inequality sign should be placed to the left of the number.

Reviewer 2 Report

Thank you for the opportunity to review this manuscript. It is study addressing the secondary attack rare of COVID-19 among non-spousal household contacts in Japan. All information regarding the transmission of covid is of interest and so I feel this paper has value in being published.

This is a solid paper and is very succinct which is appropriate.

I have one major issue to be addressed.

The major issue is that there is no mention of different variants in this paper, either in the introduction or discussion and this is potentially a significant part of the story, and perhaps a limitation of the study if variants have different transmission characteristics and you have multiple variants that are all mixed in together. I think this issue needs to be addressed in this paper and cannot be ignored.

I have a couple of minor issues.

1. I think the results section of the abstract should be re-written as it is just too hard to make sense of in the way it is currently written unless you have read the paper already. I think the results can be made clearer in the abstract and this should be completed.

2. One possible limitation is that index cases could have been misclassified as secondary cases, even taking into account the time of onset of infection, isn’t it? I think this needs to be discussed if it is a possibility, even if it is unlikely.

Reviewer 3 Report

1) Introduction is superficial, please provide a more deep context and explanation of the situation. For example, the evidence of how the SARS-CoV-2 is transmitted, the level of contagious, and all the factos that can increase or decrease transmission. Also provide in the introduction a more deep context about Japan's SARS-CoV-2 outbreak.

2) The methodology of the study is adequate and very nice presented. 

3) In table 1. which has the median or mean (depending data distribution) the of the number of households, diagnostic delay and age, also please add the adequate dispersion measure value. Please evaluate this variable as a continuous not only as categorical. 

4) The results from line 155-158 are very relevante, please not only describe the result, provide a deeper analysis of this result. 

5) In table 2, 3 and 4 add to the aOR the 95% CI.

6) in the discussion please provide a deeper explanation about the importance and soundness of your results. 
